# eHealth literacy and its associated factors in Ethiopia: Systematic review and meta-analysis

Sisay Maru Wubante[1,2,3]*, Masresha Derese Tegegne[1,2,3], Mequannent
Sharew Melaku[1,2,3], Mulugeta Hayelom Kalayou[4], Yeshambel Andargie Tarekegn[1,2,3],
Sintayehu Simie Tsega[1,2,3], Nebyu Demeke Mengestie[1,2,3], Addisalem Workie Demsash[5],
Agmasie Damtew Walle[5]

**1** Department of Health Informatics, Institute of Public Health, College of Medicine and Health Sciences,
University of Gondar, Gondar, Ethiopia, **2** Department of Medical Nursing, School of Nursing, College of
Medicine and Health Science, University of Gondar, Gondar, Ethiopia, **3** Department of Otorhinolaryngology
(ENT), School of Medicine, College of Medicine and Health Science, University of Gondar, Gondar, Ethiopia,
**4** Department of Health Informatics, Institute of Public Health, College of Medicine and Health Sciences,
Wollo University, Wollo, Ethiopia, **5** Department of Health Informatics, Institute of Public Health, College of
Medicine and Health Sciences, Mettu University, Mettu, Ethiopia

* sisay419@gmail.com

Medical Sciences, ISLAMIC REPUBLIC OF IRAN

**Data Availability Statement:** All relevant data are
within the paper and its Supporting Information
files.

**Funding:** There is no funding for this study.

## Abstract

### Introduction

Electronic health has the potential benefit to the health system by improving health service
quality efficiency effectiveness and reducing the cost of care. Having good e-health literacy
level is considered essential for improving healthcare delivery and quality of care as well as
empowers caregivers and patients to influence control care decisions. Many studies have
done on eHealth literacy and its determinants among adults, however, inconsistent findings
from those studies were found. Therefore, this study was conducted to determine the pooled
magnitude of eHealth literacy and to identify associated factors among adults in Ethiopia
through systematic review and meta-analysis.

### Method

Search of PubMed, Scopus, and web of science, and Google Scholar was conducted to find
out relevant articles published from January 2028 to 2022. The Newcastle-Ottawa scale tool
was used to assess the quality of included studies. Two reviewers extracted the data inde-
pendently by using standard extraction formats and exported in to Stata version11 for meta-
analysis. The degree of heterogeneity between studies was measured using I2 statistics.
The publication bias between studies also checked by using egger test. The pooled magni-
tude of eHealth literacy was performed using fixed effect model.

### Result

After go through 138 studies, five studies with total participants of 1758 were included in this
systematic review and Meta-analysis. The pooled estimate of eHealth literacy in Ethiopia
was found 59.39% (95%CI: 47.10–71.68). Perceived usefulness (AOR = 2.46; 95% CI:
1.36, 3.12),educational status(AOR = 2.28; 95% CI: 1.11, 4.68), internet access (AOR =

**Competing interests:** The authors have declared that no competing interests exist.

**Abbreviations:** WHO, World Health Organization; Ehealth, electronic health; AOR, adjusted odds ratio; CI, confidence interval.

2.35; 95% CI: 1.67, 3.30), knowledge on electronic health information sources(AOR = 2.60; 95% CI: 1.78, 3.78), electronic health information sources utilization (AOR = 2.55; 95%CI: 1.85, 3.52), gender (AOR = 1.82; 95% CI: 1.38, 2.41) were identified significant predictors of e-health literacy.

## Conclusion and recommendation

This systematic review and meta-analysis found that more than half of study participants were eHealth literate. This finding recommends that creating awareness about importance of eHealth usefulness and capacity building to enhance and encouraging to use electronic sources and availability of internet has para amount to solution to increase eHealth literacy level of study participants.

## Introduction

Health professionals and patients can be communicated through the eHealth platform to make evidence-based decisions and share health information about health status [1]. Today electronic solutions are increasingly used as key means of communication and looking for Nobel health information between the healthcare provider and patients and this term is known as eHealth [2].

According to the World Health Organization (WHO), eHealth can be defined as the application of the Internet and other related technologies in the healthcare industry to improve the access, efficiency, effectiveness, and quality of clinical and business processes utilized by healthcare organizations, practitioners, patients, and consumers to improve the health status of patients [3, 4]. Ehealth literacy is the capacity to analyze health information obtained from electronic sources and apply what is learned to address or solve a health problem [1, 4, 5].

Studies show that e-health literacy is considered essential for improving healthcare delivery and quality of care as well as empowering caregivers and patients to influence control care decisions [2–4, 6]. Although eHealth has the potential to revolutionize medical and public health practice, several organizational, cultural, and human resource reforms are still required for the widespread adoption of eHealth techniques for finding high-quality health information [3].

According to a study conducted in Ethiopia, considering the country's low internet penetration rate (15%), finding and evaluating online information is still a challenge [6]. Another cross-sectional study showed that 72.6% of the variation in eHealth literacy is explained [7]. Students studying medicine and health sciences in Ethiopia have a very low level of eHealth literacy, which leads to health disparities that encourage the development of chronic diseases and greater healthcare expenses that harm patient outcomes [8].

Poor eHealth literacy has been associated with numerous adverse health outcomes, according to various studies, including trouble navigating the healthcare system, inaccurate or sparse medical history reports, missed doctor appointments, incorrect medication uses in terms of timing or dosage, decreased rates of adherence to chronic illness regimens, and an increased risk of hospitalization [3, 5].

According to the literature, eHealth literacy has a positive effect on improving health conditions and increasing the quality of healthcare service delivery [9]. Educational background, knowledge of electronic health information resources, internet use, perceived usefulness, electronic health information resource utilization, and gender are determinants of eHealth literacy [9, 10]. Dealing with these problems will have a significant contribution to the improvement of

the quality of health and healthcare services. Furthermore, assessing 'the eHealth literacy level and its determinants would allow the government to identify eHealth literacy levels and impediments to designing appropriate plans. Policymakers and other stakeholders will benefit from the findings of this systematic review and meta-analysis as input for implementing various digital health implementation initiatives to overcome barriers to primary health coverage in the country. One of the most important factors driving the importance of assessing eHealth literacy and its determinants to capitalize on eHealth benefits is the growing digital health penetration in African countries. Given the low number of systematic reviews and meta-analysis studies on eHealth literacy in Africa, this research will contribute to scientific knowledge by addressing the research question in this region. The finding of this systematic review and meta-analysis will have baseline information for future researchers to conduct interventional studies regarding eHealth literacy. Therefore this study aimed to assess eHealth literacy and its associated factors systematic review and meta-analysis in Ethiopia.

## Materials and methods

This systematic review and meta-analysis are conducted accordingly to Preferred Reporting Items for Systematic Reviews and Meta-Analysis (PRISMA) checklist. This systematic review and meta-analysis are registered with protocol number CRD42022340469.

### Eligibility criteria

Primary studies reporting eHealth literacy and its associated factors among, students, health professionals, and patients in Ethiopia were included in this systematic review meta-analysis. Studies conducted from January 2018 up to January 2022 were included. Studies did not clearly show that ehealth literacy and its associated factors in Ethiopia were excluded. In addition abstracts, articles without full text, and grey and unpublished works were excluded.

**Search strategy and information sources.** A compressive and systematic way of retrieving articles was conducted through ONLINE databases including PubMed, Embrace, Web of Science, Scopus, Google scholar, and the African journal online was undertaken from January 2018 and January 2022 to find relevant published articles. The retrieval was conducted using the following keywords and Medical Subject Headings (MeSH) terms: "e-health literacy" (or electronic health literacy OR digital literacy) AND (health professionals OR patients OR students) AND Ethiopia". The retrieval focuses on eHealth literacy and its associated factors among students, health professionals, and patients in Ethiopia.

### Data extraction

After a careful selection of articles included in this study, Data were extracted using a standardized data extraction tool adopted from Joanna Briggs Institute. Two independent reviewers (SMW&ADW) extracted the data and review the whole article. The following study characteristics were such as the first author's name, year of publication, number of participants, participants' characteristics and setting, sample size, data collection techniques, and study design were included. The magnitude of Ehealth literacy and associated factors, with 95% confidence intervals, were also extracted.

### Assessment of the risk of bias

The Newcastle-Ottawa scale (NOS) tool was designed for cross-sectional study quality assessments to measure the quality of each original study. The assessment tool is divided into three sections: The tool's first section is a five-star rating system that evaluates each study's

methodological quality (i.e., sampling technique, sample size, and ascertainment of the associated factor). The tool's second section evaluates the study's comparability, with the option of obtaining two stars. The final component of the instrument assesses the primary study's outcomes and statistical tests, with the possibility of earning three stars. Finally, the quality of the studies included in this systematic review and meta-analysis ranges from (5–6 out of 10) to good, to high-quality ratings (> 6 stars out of 10) the quality of the papers included in the review was appraised separately by two writers. During the quality evaluation, disagreements between reviewers were resolved through communication.

## Statistical analysis

A Microsoft Excel spreadsheet was used to extract the data, which was then imported into STATA version 11 for further analysis. Tables, figures, and forest plots were used to describe and summarize the main study. Analyses were carried out to determine the associations between e-health literacy and associated factors. The effect size was calculated using the pooled odds ratio from studies that reported odds ratios. Meta-analyses were carried out, and forest plots were made. The degree of heterogeneity between studies was measured using I2 statistics. I2 represented heterogeneity levels with values of 25%, 50%, and 75%, indicating low, moderate, and high heterogeneity, respectively. We used a random-effect model to allow for high heterogeneity because the studies included differed in terms of participants, settings, and measurement. We used Egger's test to test for publication bias in the meta-analysis, and the results were illustrated in a funnel plot.

## Result

### Search result

Initially, 138 records were found through database searching about e-health literacy and determinants. After duplicates were excluded, 25 abstracts were reviewed, and five were selected for further consideration. Primarily due to differences in statistical regression and outcome measurement across more than two categories. As a result, five papers were chosen to participate in the final systematic review and meta-analysis (Fig 1).

### Characteristics of the included studies

To estimate the eHealth literacy magnitude, all of the included research used a facility-based cross-sectional study design was employed. All of the research analyzed in this systematic review and meta-analysis was published between 2018 and 2022. Three of the studies in this review employed stratified sampling, whereas two used simple random sampling. Four of the included studies employed self-administered and one interviewer-administered methods to choose study participants. About 1758 participants took part in the survey. According to the research examined, e-health literacy ranged from 46.5% to 69.3% (Table 1).

**The pooled magnitude of e-health literacy in Ethiopia.** The five studies conducted in Ethiopia explored different populations, including chronic patients and health professionals. All of them found that the level of e-health literacy in Ethiopia is moderate According to the findings of this systematic review and meta-analysis, the pooled magnitude of e-health literacy in Ethiopia was 59.39% (95%CI: 47.10–71.68). Hereafter to estimate the pooled magnitude on Ehealth literacy random-effects model was applied during the meta-analysis. The random-effects model output showed that there is no heterogeneity between each primary study with (I2 = 0.0%; p > 0.850). Therefore no further subgroup analysis needed to be performed (Fig 2).

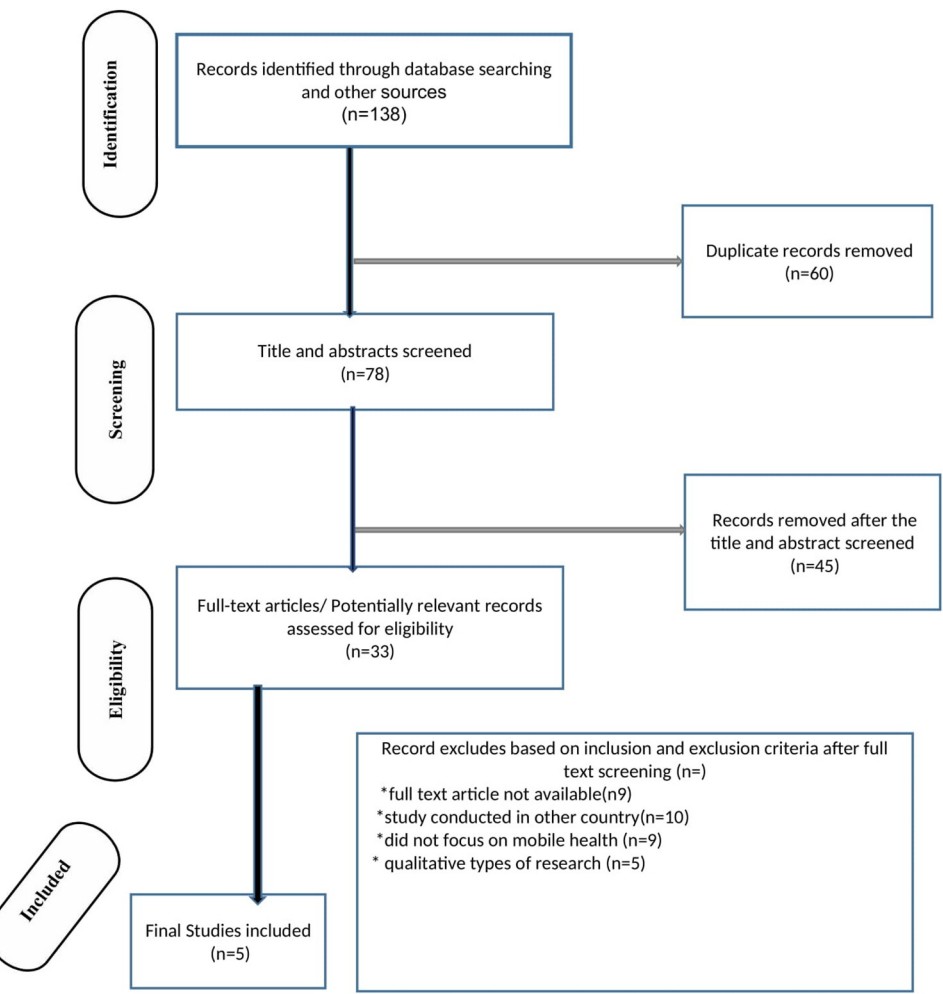

**Fig 1. Diagram of study selection for systematic review and meta-analysis of eHealth literacy and associated factors in Ethiopia, 2022.**

**Publication bias.** A graphic review of the asymmetry in a funnel plot and an Egger's regression test was used to determine the presence or absence of publication bias. Accordingly, the results of the funnel plots and Egger's regression test in this meta-analysis revealed the

**Table 1. Characteristics of individual studies done on eHealth literacy in Ethiopia 2022.**

| Author | Year of Publication | sampling technique | data collection technique | study area | Study Year | Study design | study population | Sample Size | Magnitude |
|---|---|---|---|---|---|---|---|---|---|
| Menagerie et al. | 2021 | Stratified sampling | self-administered | university of Gondar | 2019 | institutional-based cross-section | students | 801 | 60.01 |
| eden.et.al | 2019 | simple random | self-administered | university of Gondar | 2018 | institutional based cross-section | health professionals | 291 | 69.3 |
| fikadie.et.al | 2020 | Stratified sampling | interviewer administered | university of Gondar | 2020 | institutional-based cross-section | chronic patients | 423 | 46.5 |
| Shiferaw.et.al | 2020 | Stratified sampling | self-administered | Northwest | 2019 | institutional based cross section | nursing students | 236 | 60.7 |
| tesfa.et.al | 2022 | simple random | self-administered | Northwest | 2020 | institutional based cross section | health professionals | 423 | 58.7 |

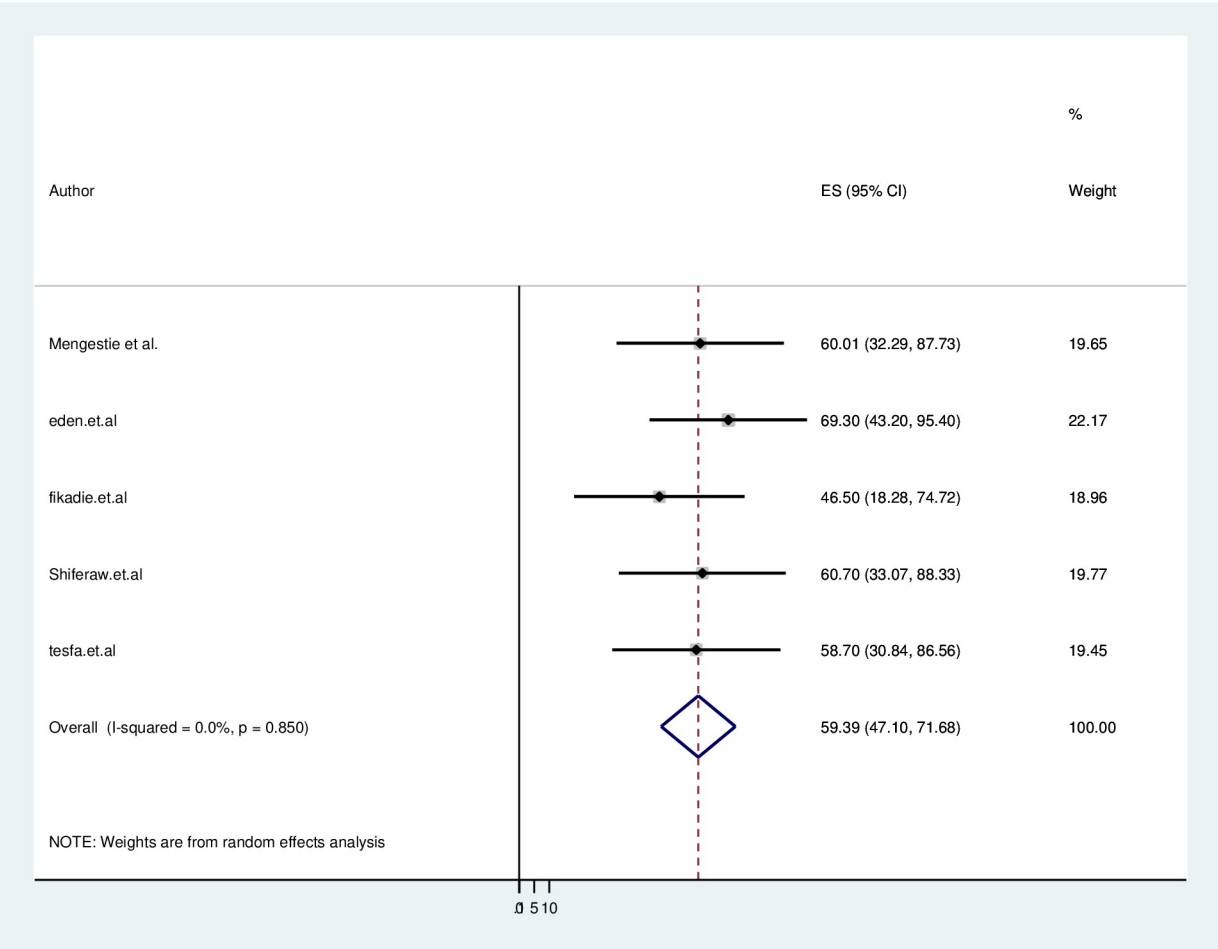

**Fig 2. Forest plot showing the pooled prevalence of eHealth literacy in Ethiopia in 2022.**

absence of publication bias in included studies. symmetrical distribution is evidence of the no presence of publishing bias, and visual inspection of the funnel plot also reveals asymmetry distribution. Similarly, Egger's test resulted in a statistically significant result for the absence of publication bias (Fig 3).

**e-health literacy and its determinants.** Systematic review and meta-analysis study, there were some predictor variables associated with e-health literacy (Fig 4).

Two studies showed that perceived usefulness has a significant association with e-health literacy the odds of Ehealth literacy were 2.46 times (AOR = 2.46; 95% CI: 1.36, 3.12) higher among respondents than their counterparts.

Two reports showed that educational status has positively associated with e-health literacy. Study participants who had a degree and above educational status were 2.28 times (AOR = 2.28; 95% CI: 1.11, 4.68) more likely to be e-health literate than their equivalents.

Two studies showed that internet access has a positive correlation with Ehealth literacy. The odds of e-health literacy were 2.35 times (AOR = 2.35; 95% CI: 1.67, 3.30) higher among study subjects who have internet access when compared with those who did not have internet access.

Two studies also showed that knowledge of electronic health information resources has an association with Ehealth literacy. The odds of Ehealth literacy were about 2.60 times

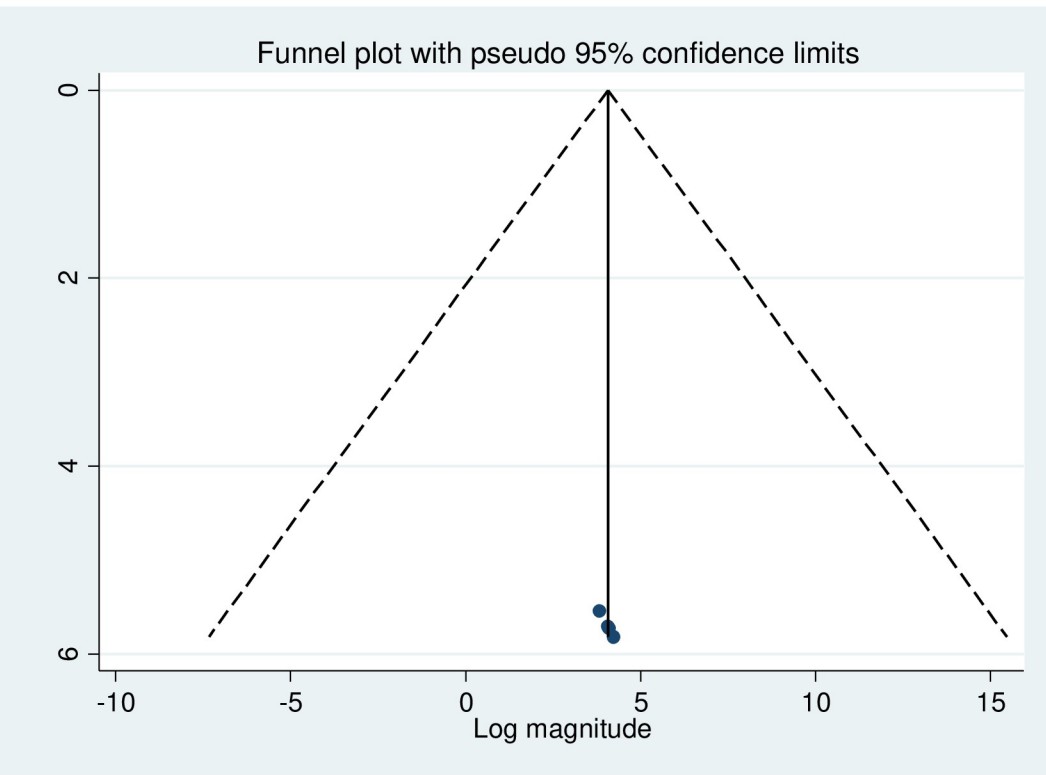

**Fig 3. Funnel plot for detecting publication bias on eHealth literacy in Ethiopia 2022.**

(AOR = 2.60; 95% CI: 1.78, 3.78) higher among respondents who have good knowledge than their counterpart.

Two studies indicated that electronic health information resource utilization has an association with Ehealth literacy. The odds of Ehealth literacy were about 2.55 times (AOR = 2.55; 95%CI: 1.85, 3.52) higher among study participants who utilized electronic health information resources than their counterparts.

Two studies showed that gender has a significant association with e-health literacy. The odds of Ehealth literacy were 1.82 times (AOR = 1.82; 95% CI: 1.38, 2.41) higher among female study subjects when compared to males.

## Discussion

With both the improvement of information technology and the growing accessibility of electronic health information, the ease of accessing and using health information has now become crucial. To the best of our knowledge, there is no systematic review and meta-analysis study done to estimate the pooled magnitude of e-health literacy and its predictors in Ethiopia. Therefore this systematic review and meta-analysis aimed to generate the pooled magnitude of e-health literacy and its predictors in Ethiopia.

The result showed that (49.39%) of participants had good eHealth literacy. This result is consistent with findings conducted in other countries [11–13]. When compared to studies conducted in other countries, the eHealth literacy of adults in this study was relatively lower [14–18]. This might be due to less internet penetration and low economic barriers in resource-limited settings like Ethiopia, low information communication technology utilization in low-income settings, and poor accessibility of health information can affect the eHealth literacy

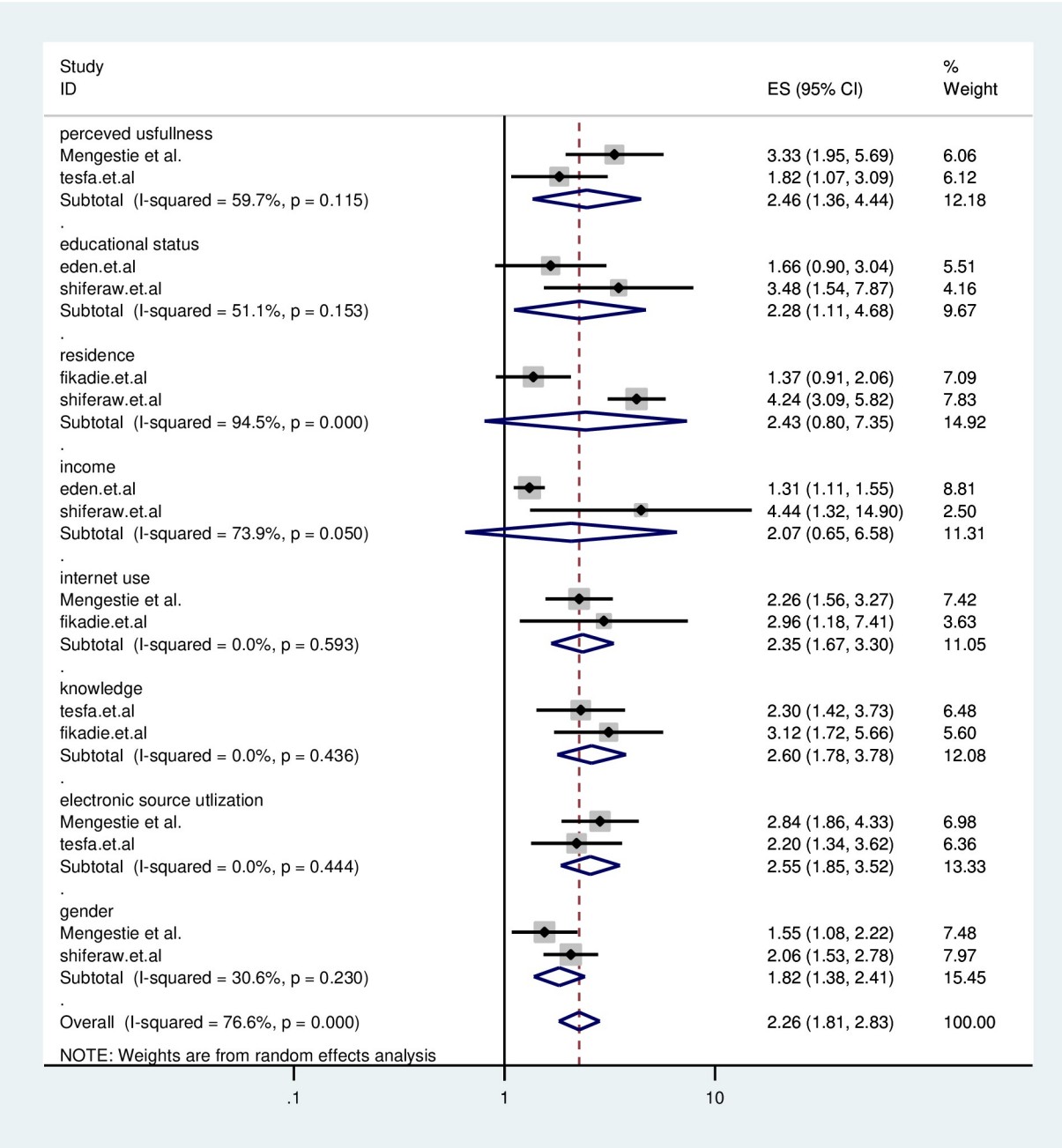

**Fig 4. Graphical representation of factors associated with eHealth literacy in Ethiopia 2022.**

level. In Ethiopia, awareness and utilization of electronic technologies among communities are low which might have influenced their eHealth literacy level. Furthermore, because of the scarcity of health-related information in languages other than English, geographical, cultural, and language issues may influence e-health. Despite the increased availability and acceptance of eHealth information and communication, all populations, including health professionals, may lack the skills to keep up with this dynamic and changing medium.

In this study perceived usefulness was found positively associated with the eHealth literacy level of study participants. Participants who thought eHealth information resources were useful for making decisions were more eHealth literate than those who thought they weren't. This result is in line with studies conducted in other countries [19–21]. It has been proposed that eHealth be integrated into the health care system because it has the potential to improve the quality of health care received [22]. Health professionals must be up to date on the latest information and skills to be competent in using eHealth resources for patient care and clinical decision-making. According to a preliminary situation assessment, Ethiopian eHealth initiatives are small in scale and unable to effectively communicate with one another (low interoperability). As a result, the Ethiopian government developed and implemented a national eHealth strategy to coordinate and streamline the country's eHealth initiatives and lay the groundwork for long-term eHealth implementation [23].

This study also identified that internet access was another determinant of eHealth literacy. The odds of having high eHealth literacy are about 2.35 times higher among adults with good Internet access compared with those with poor Internet access. This finding is consistent with studies done in [24–27]. This might be because the respondents confirmed that the internet aided them in attempting to make health-related decisions, and they assumed they knew where to find and how to use helpful health resources on the internet. In addition, Participants were sure of their ability to evaluate the information they had obtained. The participant's ability to differentiate between high- and low-quality health resources on the internet.

Electronic health source utilization has been positively associated with eHealth literacy. study participants who utilize electronic health information sources were more likely eHealth literate as compared with those not utilized. This finding is consistent with the study done in another country [28]. This might be because prior experiences in using electronic resource utilization will impact the eHealth literacy level of individuals.

Having knowledge of electronic health information sources has significantly associated with eHealth literacy. Respondents who have good knowledge of electronic health information sources were more likely to have eHealth literacy as compared to those with poor knowledge. This finding is in line with other studies conducted in [29–31]. This could have a prior understanding of electronic health information resources will increase the eHealth literacy level of individuals.

Furthermore, gender was found significantly associated with eHealth literacy. The odds of eHealth literacy are higher among females than males. This finding is consistent with findings reported in [32].

## Limitation

Even though the first systematic review and meta-analysis was conducted to evaluate the eHealth literacy level in Ethiopia it has its limitations. This systematic review and meta-analysis include only five full-text articles and were published in the English language. Does not include studies having different categorizations of the variables. In addition, only institutional-based cross-sectional studies were involved in this systematic review and meta-analysis. Future researchers better focused on eHealth literacy levels across the global region.

## Conclusion and recommendation

This systematic review and meta-analysis showed that pooled estimate of eHealth literacy was found good. Knowledge of electronic health information sources, perceived usefulness, gender, knowledge of electronic health information sources, electronic health information source utilization, and internet access were positively associated with eHealth literacy level. This study

suggested that creating awareness and motivation about the importance of digital information and its utilization is a very mandatory solution to improving Ehealth literacy levels. Improving the accessibility and availability of internet penetration in the country also has a paramount of significance on scaling up eHealth literacy.

## Supporting information

**S1 Checklist.**
(DOCX)

**S1 Data.**
(XLSX)

## Acknowledgments

We would like to thank all authors of the studies included in this systematic review and meta-analysis

## Author Contributions

**Conceptualization:** Sisay Maru Wubante, Masresha Derese Tegegne.

**Data curation:** Sisay Maru Wubante, Masresha Derese Tegegne, Yeshambel Andargie Tarekegn.

**Formal analysis:** Sisay Maru Wubante, Yeshambel Andargie Tarekegn, Agmasie Damtew Walle.

**Funding acquisition:** Sisay Maru Wubante.

**Investigation:** Sisay Maru Wubante, Nebyu Demeke Mengestie.

**Methodology:** Sisay Maru Wubante, Mulugeta Hayelom Kalayou.

**Project administration:** Sisay Maru Wubante.

**Resources:** Sisay Maru Wubante.

**Visualization:** Sintayehu Simie Tsega.

**Writing – original draft:** Mequannent Sharew Melaku.

**Writing – review & editing:** Sintayehu Simie Tsega, Nebyu Demeke Mengestie, Addisalem Workie Demsash.

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
