## [Decision Letter · Decision Letter 0]

24 Oct 2022

PONE-D-22-21015eHealth Literacy and its associated factors in Ethiopia: systematic review and meta-analysis.PLOS ONE

Dear Dr. Sisay wubante maru,

Thank you for submitting your manuscript to PLOS ONE. After careful consideration, we feel that it has merit but does not fully meet PLOS ONE’s publication criteria as it currently stands. Therefore, we invite you to submit a revised version of the manuscript that addresses the points raised during the review process.

We look forward to receiving your revised manuscript.

Kind regards,

Jahanpour Alipour, Ph.D.

Academic Editor

PLOS ONE

Journal Requirements:

"No"

"No"

Reviewers' comments:

Reviewer's Responses to Questions

**Comments to the Author**

1. Is the manuscript technically sound, and do the data support the conclusions?

Reviewer #1: No

Reviewer #2: Partly

Reviewer #3: Yes

2. Has the statistical analysis been performed appropriately and rigorously? 

Reviewer #1: No

Reviewer #2: Yes

Reviewer #3: Yes

3. Have the authors made all data underlying the findings in their manuscript fully available?

Reviewer #1: No

Reviewer #2: Yes

Reviewer #3: Yes

4. Is the manuscript presented in an intelligible fashion and written in standard English?

Reviewer #1: No

Reviewer #2: Yes

Reviewer #3: Yes

5. Review Comments to the Author

Reviewer #1: Paper must be improved in all sections. It is not clear your statistical achievements of paper. If any, it is not too evident for reader. In your abstract it was written the numerical results. A Journal paper requires in a transparent manner your exact methodology about how you got such statistical results. Finally, it is not common that a research of such importance as ehealth systems at Ethiopia have only 47 words as conclusion.

Reviewer #2: Heterogeneity test is the homogeneity test of statistics, which aims to check whether there is heterogeneity in the results of independent studies. If there is no or low heterogeneity among the studies (P＜0.1, heterogeneity level＞50%), the fixed effect model is used for data consolidation analysis; If there is heterogeneity (P ＜ 0.1, heterogeneity level ＞ 50%), but the combined data still have clinical significance, the random effect model is used for combined analysis.

Reviewer #3: Reviewer comments for PLOS ONE_PONE-D-22-21015

Title of the manuscript: eHealth Literacy and its associated factors in Ethiopia: Systematic review and meta-analysis

Thank you for the opportunity to review a timely and interesting manuscript. I look forward to seeing the manuscript in print, and the comments below are meant to strengthen the manuscript towards this end.

The study aimed to generate the pooled magnitude of e-health literacy and its predictors among adults in Ethiopia. Much as I appreciate the enormous amount of hard work put in by the author(s) and the potential contributions of their study, I am afraid that reworking will be required before the manuscript can be considered for publication. I will outline some major issues below:

Study selection criteria:

Given the large and diverse potential audiences for this work, the author(s) might want to be more explicit about the exclusion of “Gray and unpublished artifacts were excluded.”

Discussion:

The findings mentioned some major findings, but it is not immediately evident what new findings the study has produced (or what is unique about the study?). For example:

(a) The statement “However it is significantly lower than studies conducted in (12-16). This might be due to differences in socioeconomic status, digital technology penetration, educational system, and study participants.” (So, what about it? What can be inferred from this finding or possibly the differences mentioned? Signify? Implied?)

(b) The statement “The same findings were obtained suggesting a strong correlation between the perceived usefulness of the Internet to make health decisions and the level of eHealth literacy (17-19).this is due to knowing about internet users will have an impact on increasing eHealth literacy of individuals. (Please re-write for clarity)

(c) The statement “This finding is consistent with studies done in (20-23).this could be having internet access will enforce the individual to search about digital technologies applications on health.” (Apart from consistent findings with other studies, what can you imply or interpret from it? and How?)

Much of this section is merely descriptive with some interpretation. Critical interpretation and connection are missing. The author(s) might want to be more explicit about the significant contribution that the study made to the advancement of knowledge and understanding in the chosen area of study. Overall, I believe the discussion needs to/can be strengthened to highlight more explicitly the link between the relevant factors of e-health literacy.

Limitation:

6. There should be a critical reflection on the limitations of the study (systematic review and meta-analysis) before the recommendations. In the current form, the limitation lacks criticality and interpretation of the impact.

Conclusion:

7. A conclusion is presented in the manuscript but is insufficient. The question of ‘where do we go from here?’ is not presented to the reader(s). The author(s) may consider succinctly informing the reader(s) how and why it is that what has been presented in this manuscript is significant for policy, practice, or further research or action. The conclusion section should/must also address the implications.

6. PLOS authors have the option to publish the peer review history of their article (what does this mean?). If published, this will include your full peer review and any attached files.

Reviewer #1: No

Reviewer #2: No

Reviewer #3: No

---

## [Author Response · Author response to Decision Letter 0]

14 Dec 2022

Dear Editors of PLOS ONE :

It has been recalled that we the authors of the manuscript entitled “eHealth Literacy and its associated factors in Ethiopia: Systematic Review and Meta-Analysis” submitted our manuscript for publication in your journal and received reviewer comments for the betterment of the manuscript before its publication. In line with this, all authors are very happy with the constructive and valuable comments given by reviewers. Accordingly, we have considered all the comments and provided a point-by-point response and explanations for all the questions raised. Finally, we have submitted all the required documents in their revised form. We hope that we have addressed all the questions and if you have any points for further clarity, let us know.

All the authors would like to thank the editorial team and reviewers

Editor(s)’ comments to the authors 

Comment1: A rebuttal letter that responds to each point raised by the academic editor and reviewer(s). You should upload this letter as a separate file labeled 'Response to Reviewers'.

Answer: Thanks dear editor for your nice comments and suggestions. We, the authors

of this study, have attached the necessary files and a detailed rebuttal letter according to

your suggestion and the journal format.

Comment2: A marked-up copy of your manuscript that highlights changes made to the original version. You should upload this as a separate file labeled 'Revised Manuscript with Track Changes.

Answer: Thank you, the track changes and cleaned document have been prepared and labeled as revised manuscript and attached

Comment 3: An unmarked version of your revised paper without tracked changes. You should upload this as a separate file labeled 'Manuscript'.

Answer: thank you very much, dear editor, unmarked version of the revised manuscript was prepared without track change labeled as the manuscript was uploaded.

Journal Requirements:

 Comments: Please review your reference list to ensure that it is complete and correct. If you have cited papers that have been retracted, please include the rationale for doing so in the manuscript text, or remove these references and replace them with relevant current references. Any changes to the reference list should be mentioned in the rebuttal letter that accompanies your revised manuscript. If you need to cite a retracted article, indicate the article’s retracted status in the References list and also include a citation and full reference for the retraction notice.

Answer: thank you dear editor we authors of this study agreed with your comments and suggestions. We tried to review the whole references cited in the manuscript are complete and correct. we ensured that all references are correct and complete no retracted papers are cited in our manuscript.

Reviewer 1 comment to the author

Comment#reviwer1: Paper must be improved in all sections. It is not clear your statistical achievements of the paper. If any, it is not too evident to the reader. In your abstract, it was written the numerical results. A Journal paper requires transparently your exact methodology about how you got such statistical results. Finally, it is not common that research of such importance as ehealth systems in Ethiopia have only 47 words as a conclusion.

Answer: thank you very much for your valuable comments ad suggestion for the improvement of our manuscript, we authors are happy to accept and agreed with the comments. Accordingly, we revised all sections as recommended in the main revised manuscript. The abstract section is written clearly and in an understandable manner. The method, result, and discussion section are refined again in the revised manuscript.

Comment Reviewer #2: The heterogeneity test is the homogeneity test of statistics, which aims to check whether there is heterogeneity in the results of independent studies. If there is no or low heterogeneity among the studies (P＜0.1, heterogeneity level＞50%), the fixed effect model is used for data consolidation analysis; If there is heterogeneity (P ＜ 0.1, heterogeneity level ＞ 50%), but the combined data still have clinical significance, the random effect model is used for combined analysis

Answer#reviwer2: thank you very much dear reviewer for your insightful, fruitful, and constructive comments to improve our manuscript document development. We authors of this study gladly agreed and accepted all your concerns. Hence the result of this meta-analysis showed that there is no heterogeneity between individual articles, we employed fixed effect model analysis. However, in the submitted paper we say that the random effect model was used, it was wrongly written and corrected in the revised manuscript.

Comment Reviewer #3: Title of the manuscript: eHealth Literacy and its associated factors in Ethiopia: Systematic review and meta-analysis Thank you for the opportunity to review a timely and interesting manuscript. I look forward to seeing the manuscript in print, and the comments below are meant to strengthen the manuscript towards this end. The study aimed to generate the pooled magnitude of e-health literacy and its predictors among adults in Ethiopia. Much as I appreciate the enormous amount of hard work put in by the author(s) and the potential contributions of their study, I am afraid that reworking will be required before the manuscript can be considered for publication. I will outline some major issues below:

Answer Reviewer #3:: thank you very much, dear reviewer, for your appreciation of our work and for giving constructive comments on our paper's advancement. We authors of this study gladly accepted and agreed with your comments. Accordingly, refined all sections in the revised manuscript.

Comment Reviewer #3: Given the large and diverse potential audiences for this work, the author(s) might want to be more explicit about the exclusion of “Gray and unpublished artifacts were excluded.”

Answer Reviewer #3: thank you so much for your valuable contribution to improving this manuscript document, we are happy to accept your interesting comment. Since gray and unpublished papers are not passed through the previewed process the information gained from them is not helpful for decision-making and policy development. the uncertainty of the status of this information generated from such articles. 

The findings mentioned some major findings, but it is not immediately evident what new findings the study has produced (or what is unique about the study?). For example:

Comment Reviewer #3 (a): The statement “However it is significantly lower than studies conducted in (12-16). This might be due to differences in socioeconomic status, digital technology penetration, educational system, and study participants.” (So, what about it? What can be inferred from this finding or possibly the differences mentioned? Signify? Implied?)

Answer Reviewer #3: thank you dear reviewer for your interesting comments and suggestion forwarded to our manuscript advancement. Accordingly, we tried to show the inference of the finding in a more compressive way in the revised manuscript

Comment Reviewer #3 (b)The statement “The same findings were obtained suggesting a strong correlation between the perceived usefulness of the Internet to make health decisions and the level of eHealth literacy (17-19).this is due to knowing about internet users will have an impact on increasing eHealth literacy of individuals. (Please re-write for clarity)

Answer Reviewer #3: thank you so much for the nice comments, we authors of this study gladly accept and agree with this comment. Accordingly, we wrote the above reason in a clear way that the reader can understand in the revised manuscript.

Comment Reviewer #3 (c) The statement “This finding is consistent with studies done in (20-23).this could be having internet access will enforce the individual to search about digital technologies applications on health.” (Apart from consistent findings with other studies, what can you imply or interpret from it? and How?) 

Answer Reviewer #3: thank you very much, dear reviewer, we are happy to accept your comments on the improvement of our manuscript. accordingly, we tried to show interpreting reasons rather than listing them in the revised manuscript.

Comment Reviewer #3: Much of this section is merely descriptive with some interpretation. Critical interpretation and connection are missing. The author(s) might want to be more explicit about the significant contribution that the study made to the advancement of knowledge and understanding in the chosen area of study. Overall, I believe the discussion needs to/can be strengthened to highlight more explicitly the link between the relevant factors of e-health literacy.

Answer Reviewer #3: thank you, we author of this study accepted and agreed on your recommended comments to our manuscript suite for publication. Accordgly, we authors refined more this manuscript by showing the significant contribution to generating knowledge on the area. The discussion part strengthens and tries to show factors associated with eHealth literacy level.

Limitation:

Comment Reviewer #3: There should be a critical reflection on the limitations of the study (systematic review and meta-analysis) before the recommendations. In the current form, the limitation lacks criticality and interpretation of the impact.

Answer Reviewer #3: thank you so much for your insightful comments on this manuscript, we authors of this study agreed on that comment. Accordingly, we showed the limitation of this systematic review and meta-analysis study in a more comprehensive manner to show clear evidence in the revised manuscript.

Conclusion:

Comment Reviewer #3. A conclusion is presented in the manuscript but is insufficient. The question of ‘where do we go from here?’ is not presented to the reader(s). The author(s) may consider succinctly informing the reader(s) how and why it is that what has been presented in this manuscript is significant for policy, practice, or further research or action. The conclusion section should/must also address the implications.

Answer Reviewer #3: thank you dear reviewer for your interesting comments forwarded to our manuscript development. So, we tried to show the conclusion section of the manuscript more about the implication of the finding and future directions in the revised manuscript

---

## [Decision Letter · Decision Letter 1]

20 Jan 2023

PONE-D-22-21015R1eHealth Literacy and its associated factors in Ethiopia: systematic review and meta-analysis.PLOS ONE

Dear Dr. Sisay Maru Wubante,

Thank you for submitting your manuscript to PLOS ONE. After careful consideration, we feel that it has merit but does not fully meet PLOS ONE’s publication criteria as it currently stands. Therefore, we invite you to submit a revised version of the manuscript that addresses the points raised during the review process.

We look forward to receiving your revised manuscript.

Kind regards,

Jahanpour Alipour, Ph.D.

Academic Editor

PLOS ONE

Journal Requirements:

Reviewers' comments:

Reviewer's Responses to Questions

**Comments to the Author**

1. If the authors have adequately addressed your comments raised in a previous round of review and you feel that this manuscript is now acceptable for publication, you may indicate that here to bypass the “Comments to the Author” section, enter your conflict of interest statement in the “Confidential to Editor” section, and submit your "Accept" recommendation.

Reviewer #2: All comments have been addressed

Reviewer #3: (No Response)

2. Is the manuscript technically sound, and do the data support the conclusions?

Reviewer #2: Yes

Reviewer #3: Yes

3. Has the statistical analysis been performed appropriately and rigorously? 

Reviewer #2: Yes

Reviewer #3: N/A

4. Have the authors made all data underlying the findings in their manuscript fully available?

Reviewer #2: Yes

Reviewer #3: Yes

5. Is the manuscript presented in an intelligible fashion and written in standard English?

Reviewer #2: Yes

Reviewer #3: Yes

6. Review Comments to the Author

Reviewer #2: 1.This review only included 5 articles for analysis, is it too little?

2.Among the five articles included in this review, almost all the research objects are related to medical field, but the conclusion is the overall e-health literacy of Ethiopian adults. Is it unreasonable?

Reviewer #3: This is an interesting study, and I can see that efforts have been made to improve the work and address the various concerns raised previously. At this point, I would advise the authors to revisit these two elements and revise as needed.

1. In general, I did not find this study to make a valuable contribution to the literature or to eHealth Literacy. Please emphasize the contribution of this study.

2. An unaddressed limitation of the study. Any steps you could possibly take to address it in your study and how it could be addressed by future researchers?

7. PLOS authors have the option to publish the peer review history of their article (what does this mean?). If published, this will include your full peer review and any attached files.

Reviewer #2: No

Reviewer #3: No

---

## [Author Response · Author response to Decision Letter 1]

25 Jan 2023

Dear Editors of PLOS ONE :

It has been recalled that we the authors of the manuscript entitled “eHealth Literacy and its associated factors in Ethiopia: Systematic Review and Meta-Analysis” submitted our manuscript for publication in your journal and received reviewer comments for the betterment of the manuscript before its publication. In line with this, all authors are very happy with the constructive and valuable comments given by reviewers. Accordingly, we have considered all the comments and provided a point-by-point response and explanations for all the questions raised. Finally, we have submitted all the required documents in their revised form. We hope that we have addressed all the questions and if you have any points for further clarity, let us know.

All the authors would like to thank the editorial team and reviewers

Editor(s)’ comments to the authors 

Comment1: A rebuttal letter that responds to each point raised by the academic editor and reviewer(s). You should upload this letter as a separate file labeled 'Response to Reviewers'.

Answer: Thanks dear editor for your nice comments and suggestions. We, the authors

of this study, have attached the necessary files and a detailed rebuttal letter according to

your suggestion and the journal format.

Comment #2: A marked-up copy of your manuscript that highlights changes made to the original version. You should upload this as a separate file labeled 'Revised Manuscript with Track Changes.

Answer: Thank you, the track changes and cleaned document have been prepared and labeled as revised manuscript and attached

Comment 3: An unmarked version of your revised paper without tracked changes. You should upload this as a separate file labeled 'Manuscript'.

Answer: thank you very much, dear editor, unmarked version of the revised manuscript was prepared without track change labeled as the manuscript was uploaded.

Journal Requirements:

 Comments: Please review your reference list to ensure that it is complete and correct. If you have cited papers that have been retracted, please include the rationale for doing so in the manuscript text, or remove these references and replace them with relevant current references. Any changes to the reference list should be mentioned in the rebuttal letter that accompanies your revised manuscript. If you need to cite a retracted article, indicate the article’s retracted status in the References list and also include a citation and full reference for the retraction notice.

Answer: thank you dear editor we authors of this study agreed with your comments and suggestions. We tried to review the whole references cited in the manuscript are complete and correct. we ensured that all references are correct and complete no retracted papers are cited in our manuscript.

Reviewer Comment 

Comment Reviewer #2: This review only included 5 articles for analysis, is it too little?

Answer: thank you very much for your valuable comments ad suggestion for the improvement of our manuscript, we authors are happy to accept and agreed with the comments. Due to that only five articles were published regarding eHealth literacy and its determinants in the Ethiopia context. Even though five articles were included in this systematic review and meta-analysis the finding will have a high contribution to Ethiopian digital health system policy and strategy development for successful 

Comment Reviewer #2. Among the five articles included in this review, almost all the research objects are related to the medical field, but the conclusion is the overall e-health literacy of Ethiopian adults. Is it unreasonable?

Answer#reviwer2: thank you very much dear reviewer for your insightful, fruitful, and constructive comments to improve our manuscript document development. We authors of this study gladly agreed and accepted all your concerns. Hence the published articles included in this systematic review and meta-analysis showed that there are focuses in the medical field, we rewrite it again in the revised manuscript in a logical manner. 

Reviewer three comments 

Reviewer #3: This is an interesting study, and I can see that efforts have been made to improve the work and address the various concerns raised previously. At this point, I would advise the authors to revisit these two elements and revise as needed.

Answer#3: thank you so much for your appreciation of our work, we get the following comments were very constructive and have a significant contribution to our manuscript development. We act accordingly 

Reviewer #3: In general, I did not find this study to make a valuable contribution to the literature or eHealth Literacy. Please emphasize the contribution of this study.

Answer: thank you very much for your valuable comments ad suggestion for the improvement of our manuscript, we authors are happy to accept and agreed with the comments. Accordingly, we revised manuscript sections as recommended in the main revised manuscript. Accordingly, we try to put the contribution of this systematic review and meta-analysis emphasis on the eHealth literacy filed in the revised manuscript in the introduction part. 

Reviewer #3: An unaddressed limitation of the study. Any steps you could take to address it in your study and how it could address by future researchers?

 Answer #3: thank you very much for the very important comments forwarded to our manuscript betterment before publication. Accordingly, we rewrote the limitation section of this manuscript that could be addressed by future researchers.

---

## [Decision Letter · Decision Letter 2]

10 Feb 2023

eHealth Literacy and its associated factors in Ethiopia: systematic review and meta-analysis.

PONE-D-22-21015R2

Dear Dr. Sisay Maru Wubante,

We’re pleased to inform you that your manuscript has been judged scientifically suitable for publication and will be formally accepted for publication once it meets all outstanding technical requirements.

Kind regards,

Jahanpour Alipour, Ph.D.

Academic Editor

PLOS ONE

Reviewers' comments:

Reviewer's Responses to Questions

**Comments to the Author**

1. If the authors have adequately addressed your comments raised in a previous round of review and you feel that this manuscript is now acceptable for publication, you may indicate that here to bypass the “Comments to the Author” section, enter your conflict of interest statement in the “Confidential to Editor” section, and submit your "Accept" recommendation.

Reviewer #2: All comments have been addressed

Reviewer #3: All comments have been addressed

2. Is the manuscript technically sound, and do the data support the conclusions?

Reviewer #2: Yes

Reviewer #3: Partly

3. Has the statistical analysis been performed appropriately and rigorously? 

Reviewer #2: Yes

Reviewer #3: Yes

4. Have the authors made all data underlying the findings in their manuscript fully available?

Reviewer #2: Yes

Reviewer #3: (No Response)

5. Is the manuscript presented in an intelligible fashion and written in standard English?

Reviewer #2: Yes

Reviewer #3: Yes

6. Review Comments to the Author

Reviewer #2: (No Response)

Reviewer #3: Dear Authors,

After two rounds of revision and review, I felt that the authors had addressed some issues that impeded my understanding earlier on.

As a whole, the article touches on an interesting and important topic.

In terms of language use and style, some improvements can be made. There were numerous language inaccuracies throughout the article. Some parts were also repetitive. I suggest the article be proofread for language warranty before proceeding to complete acceptance of the article and publishing.

Once again, thank you for the opportunity for reviewing and for the resilience shown from all party in this process.

7. PLOS authors have the option to publish the peer review history of their article (what does this mean?). If published, this will include your full peer review and any attached files.

Reviewer #2: No

Reviewer #3: No

---

## [Editor Report · Acceptance letter]

16 Feb 2023

PONE-D-22-21015R2 

eHealth Literacy and its associated factors in Ethiopia: Systematic Review and Meta-Analysis. 

Dear Dr. Wubante:

I'm pleased to inform you that your manuscript has been deemed suitable for publication in PLOS ONE. Congratulations! Your manuscript is now with our production department. 

Kind regards, 

on behalf of

Dr., Jahanpour Alipour 

Academic Editor

PLOS ONE